# Dynamic Context Pruning for Efficient and Interpretable Autoregressive Transformers

**Sotiris Anagnostidis**[μ,*]    **Dario Pavllo**[μ,*]    **Luca Biggio**[μ,ν]    **Lorenzo Noci**[μ]

**Aurelien Lucchi**[τ]    **Thomas Hofmann**[μ]

[μ]ETH Zürich
[ν]ML, CSEM SA
[τ]University of Basel

## Abstract

Autoregressive Transformers adopted in Large Language Models (LLMs) are hard to scale to long sequences. Despite several works trying to reduce their computational cost, most of LLMs still adopt attention layers between all pairs of tokens in the sequence, thus incurring a quadratic cost. In this study, we present a novel approach that dynamically prunes contextual information while preserving the model's expressiveness, resulting in reduced memory and computational requirements during inference. Our method employs a learnable mechanism that determines which uninformative tokens can be dropped from the context at any point across the generation process. By doing so, our approach not only addresses performance concerns but also enhances interpretability, providing valuable insight into the model's decision-making process. Our technique can be applied to existing pre-trained models through a straightforward fine-tuning process, and the pruning strength can be specified by a sparsity parameter. Notably, our empirical findings demonstrate that we can effectively prune up to 80% of the context without significant performance degradation on downstream tasks, offering a valuable tool for mitigating inference costs. Our reference implementation achieves up to $2\times$ increase in inference throughput and even greater memory savings.

## 1   Introduction

The introduction of Transformers [Vaswani et al., 2017] in Large Language Models (LLMs) has profoundly influenced the landscape of Natural Language Processing (NLP), due to their appealing scaling properties [Kaplan et al., 2020] and their ability to train efficiently on modern hardware architectures designed for extensive parallel computing. As LLMs grow larger and more complex, the challenges associated with training and deploying them become more prominent. Especially challenging is the quest for processing increasingly longer sequences, as pure self-attention layers scale quadratically in sequence length during train and inference.

To address this limitation, several efforts focus on efficient implementations of the attention mechanism on dedicated hardware [Dao et al., 2022, Touvron et al., 2023], or on algorithmic procedures to directly tackle the quadratic complexity. The latter direction has led to numerous variants sacrificing the generality of the standard attention mechanism in favor of more efficient alternatives [Tay et al.,

---

[*]Equal contribution. Correspondence sotirios.anagnostidis@inf.ethz.ch.

37th Conference on Neural Information Processing Systems (NeurIPS 2023).

2020, Kitaev et al., 2020, Choromanski et al., 2020b, Katharopoulos et al., 2020, Zaheer et al., 2020, Shi et al., 2021, Lin et al., 2022, Zhu and Soricut, 2021, Dai et al., 2020], some of which are illustrated in Fig. 1. Specifically, a large number of these methods focus either on sparsifying the attention weights, reducing the size of the available context to each token, or compressing the number of tokens to reduce the size of the attention matrix.

These methods, however, are inherently static, in the sense that each token is either forced to attend to a fixed pre-specified context window, or the input context is compressed to a fixed dimensionality, regardless of the information content of the input sequence. Furthermore, a performance gap still exists with respect to pure self-attention in many applications, thus implying the existence of a non-trivial trade-off between the span of the attention context and the model's capabilities [Dao et al., 2022, Sun et al., 2021, Beltagy et al., 2020].

To address these challenges, and enhance inference efficiency, while staying faithful to pure self-attention, we pose the following question:

*Can we dynamically prune past content based on the available context,*
*while preserving as much as possible the expressivity of the model?*

In response to this question, we introduce a novel method for context pruning in Transformer-based decoder architectures. Our approach adds a minimal amount of additional training parameters that enable individual tokens to dynamically remove portions of the input sequence in a layer-wise fashion. Once part of the context is removed, it is disregarded for the remaining part of the autoregressive generation process, leading to reduced memory usage and computational requirements during inference. To this end, we also design a dynamic data structure that implements efficient insertion/removal of tokens from the context while supporting batched inference. In contrast to traditional methods relying on local or sparse attention, which may not capture the nuances and dynamic nature of the data over long contexts, ours leverages contextual cues to dynamically determine the relevance of the available information through a learned mechanism. This is achieved by making use of a sparse sigmoid function [Peters et al., 2019, Martins et al., 2020]. As demonstrated by our experimental evaluations, this allows us to extract and utilize essential details in a more adaptive and accurate manner. The degree of pruning can be effectively controlled through a hyperparameter that effectively accounts for the sparsity level.

Our technique serves as a modular building block for existing pre-trained models and can be easily integrated through a minimal fine-tuning stage. For our study, we focus on GPT-2 models [Radford et al., 2019] as they are publicly available and widely benchmarked, but due to the uniformity of modern architectures, our approach can be straightforwardly extended to any autoregressive Transformer. Moreover, since our method is based on context pruning, it can be seamlessly combined with other approaches aimed at improving inference efficiency, such as quantization, weight pruning, approximate attention, or other hardware optimizations.

We find that up to $80\%$ of the context can be successfully pruned, with minimal deterioration in terms of perplexity and zero-shot performance, while requiring significantly fewer resources during inference. We showcase how these improvements can lead to measurable practical gains, by providing an efficient implementation that reduces memory usage for caching during token generation. More specifically, for larger context sizes we get up to $50\%$ wall-time latency reduction for each generation step, while still decoding with up to $2\times$ larger batch sizes, leading thus to significant performance benefits. These findings highlight the potential of context pruning as a powerful technique to enhance the efficiency and interpretability of Transformers in NLP.

## 2 Related Work

Despite exhibiting human-level performance on a number of challenging tasks, LLMs are resource-intensive and inefficient. While the human brain consumes roughly the amount of energy equivalent to a dim light bulb, top-performing GPT models require multiple GPUs with ~80GB of memory each for inference [Strubell et al., 2019, Frantar and Alistarh, 2023a]. Several research efforts have been focusing on improving their efficiency and memory requirements from several different angles.

**Weight Pruning and Quantization.** Modern LLMs have high memory and compute requirements for both training and testing. To address this limitation, a number of research efforts [Kwon et al.,

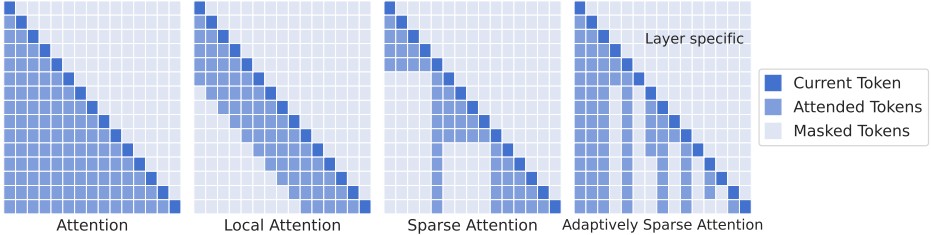

Figure 1: Visualization of the causal attention weights associated with standard, local, sparse causal attention, and our approach. Adaptively sparse attention (rightmost) prunes weights dynamically for each token, and it does not impose any restricting inductive biases on the final attention structure.

2022, Frantar et al., 2023, Frantar and Alistarh, 2023b] have resorted to the established practice of weight pruning [Hassibi et al., 1993] to efficiently compress the original model to a more manageable size. Remarkably, a large percentage of the original weights can be safely removed, resulting in only marginal perplexity growth [Bahl et al., 1983]. An alternative approach to reduce the memory and compute is quantization [Dettmers et al., 2022, Yao et al., 2022, Xiao et al., 2022, Frantar et al., 2022], which reduces the precision of the model's numerical representation. Quantization schemes [Dettmers et al., 2022] enable 8-bit matrix multiplication for both feed-forward and attention projection layers resulting in significantly improved memory allocation without incurring any performance degradation.

**Efficient Transformers and context pruning.** One primary constraint of Transformer-based models is their quadratic complexity with respect to the length of the input sequence. Extensive research explores alternatives that exhibit sub-quadratic scaling, resulting in three main strategies [Lin et al., 2022]. The first replaces the attention mechanism with an alternative operation that features more favourable scaling with the input sequence length [Peng et al., 2021, Katharopoulos et al., 2020, Choromanski et al., 2020a, Schlag et al., 2021]. While several recent methods in this category show promise, none have emerged as a definitive winner, and most state-of-the-art language models still rely on the standard attention mechanism [Touvron et al., 2023, Chowdhery et al., 2022]. The second approach proposed to compress the length of the input context, controlling the complexity of the attention operation but unavoidably sacrificing potentially relevant information from the original input [Lee et al., 2019, Wang et al., 2020, Jaegle et al., 2021]. The third approach involves pruning the attention matrix, preventing each token from attending to every other token within the context [Zaheer et al., 2020, Martins et al., 2020, Lee et al., 2023]. This line of research is motivated by the theoretical finding highlighting that sparse Transformers retain the expressivity of their dense counterparts [Yun et al., 2020]. Many methods in this category employ specially designed attention masks that aim to zero out as many entries as possible, often based on principles of locality, randomness, or a combination of both. The main drawback of these methods is their mostly static nature, meaning that every token is compelled to attend to a fixed context window and disregard the rest of the context regardless of its specific role within the input sequence. Our approach falls within this last category, and enables dynamic sparsification of the attention matrix for decoder models, without resorting to any potentially restricting inductive biases about its structure.

**Implementation Speed-up** Recently, hardware-optimized implementations [Dao et al., 2022, Touvron et al., 2023] have been proposed with the aim of optimizing computational resources during the training phase of Transformers [Hoffmann et al., 2022]. On the other hand, as recent breakthroughs have led to widespread adoption of these models [Ouyang et al., 2022, OpenAI, 2023, Köpf et al., 2023], performance during inference becomes more relevant by the day. In decoder-based autoregressive Transformers, the backbone architecture of most current state-of-the-art LLMs, inference involves evaluating and generating tokens one by one, using cached previous activations to avoid redundant computations. In contrast to training, the inference is memory bound [Shazeer, 2019, Ivanov et al., 2021, Pope et al., 2022]. Compute is under-utilized, especially when deploying larger models, as the time required to transfer model parameters and activations to hardware memory far exceeds the actual computational time. This is further exacerbated by recent trends to ever-increase the model size and enable longer context windows. As a result, batch decoding, a promising direction for more efficient utilization of hardware resources, is impeded.

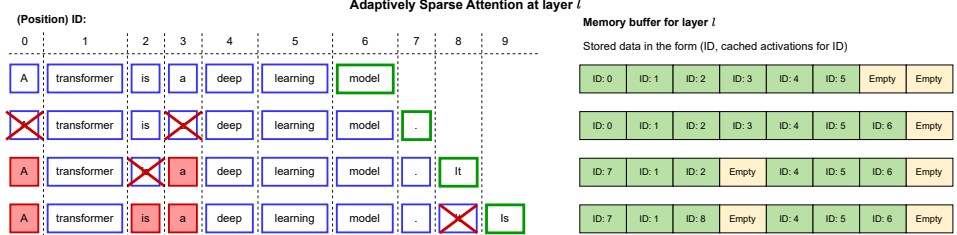

Figure 2: We illustrate the state of the memory buffer at the start of each iteration for our proposed approach. Dropped tokens are irrelevant for any subsequent generation step and their cached activations are erased. Since self-attention is a set operation, the buffer (keys/values) of the dropped tokens can be reused by subsequent tokens, ensuring that the data structure is as packed as possible. Here "X" denotes tokens that are currently being dropped. Red boxes correspond to tokens that are already dropped, for which Key-Value values are no longer being cached.

## 3 Methodology

**Background.** We operate on sequences of text tokens $\mathbf{T} \in \{0, 1, \ldots, n_{\text{vocab}}\}^n$, where $n$ is the length of the sequence and $n_{\text{vocab}}$ is the vocabulary size. Tokens are embedded into $\mathbf{X}^0 \in \mathbb{R}^{n \times d}$ using an embedding layer, where $d$ is the embedding dimension of the model. When necessary, we use the superscript $\ell \in \{1, 2, \ldots, L\}$ to denote the representations and weights at different layers. One layer of the Transformer-decoder architecture [Vaswani et al., 2017] is defined as

$$\mathbf{X} = \text{MHA}(\text{LayerNorm}(\mathbf{X}^{\ell-1})) + \mathbf{X}^{\ell-1}, \tag{1}$$

$$\mathbf{X}^\ell = \text{FF}(\text{LayerNorm}(\mathbf{X})) + \mathbf{X}, \tag{2}$$

where MHA stands for Multi-head self-attention defined as

$$\text{MHA}(\mathbf{X}) = \text{Concatenate}(\text{head}_1(\mathbf{X}), \text{head}_2(\mathbf{X}), \ldots, \text{head}_h(\mathbf{X}))\mathbf{W}_O, \quad \text{where} \tag{3}$$

$$\text{head}_i(\mathbf{X}) = \text{SA}(\mathbf{Q}_i, \mathbf{K}_i, \mathbf{V}_i). \tag{4}$$

Here $\mathbf{Q}_i = \mathbf{X}\mathbf{W}_{Q_i}$, $\mathbf{K}_i = \mathbf{X}\mathbf{W}_{K_i}$, and $\mathbf{V} = \mathbf{X}\mathbf{W}_{V_i}$ are the queries, keys and values and SA denotes the single-head self-attention. The weight matrices $\mathbf{W}_{Q_i}, \mathbf{W}_{K_i}, \mathbf{W}_{V_i} \in \mathbb{R}^{d \times p}$ linearly project the input embedding into the head dimension $p$. Finally, $\mathbf{W}_O \in \mathbb{R}^{d \times d}$ is the output projection. The feed-forward part of the Transformer is defined as

$$\text{FF}(\mathbf{X}) = \sigma_{\text{FF}}(\mathbf{X}\mathbf{W}_{F_1})\mathbf{W}_{F_2}, \tag{5}$$

where $\sigma_{\text{FF}}$ is a nonlinearity, and $\mathbf{W}_{F_1}, \mathbf{W}_{F_2}$ are linear layers with typical dimensions $\mathbf{W}_{F_1} \in \mathbb{R}^{d \times 4 \cdot d}$ and $\mathbf{W}_{F_2} \in \mathbb{R}^{4 \cdot d \times d}$. A final projection layer $\mathbf{W}_{\text{logits}} \in \mathbb{R}^{d \times n_{\text{vocab}}}$ is used to project back to the vocabulary space and predict the next token from the representations $\mathbf{X}^L$. We are focusing on Pre-LN [Xiong et al., 2020] decoder-only architectures, meaning that attention is causally masked, i.e. every input token $i$ attends to the first $i$ tokens in the input sequence. Conceptually, our method acts by predicting these attention masks using a learned mechanism in a layer-wise manner, with the introduction of additional constraints to make sure causality is preserved (i.e. if a token is dropped, it will remain dropped in the future). During inference, however, our method can efficiently be implemented by erasing tokens from the key-value cache commonly adopted in autoregressive attention models.

**Background: key-value cache.** In autoregressive Transformers, inference can be optimized by reusing pre-computed activations (keys and values) to accelerate the sequential generation of tokens [Ott et al., 2019, Vaswani et al., 2018, Wolf et al., 2020], bringing down the computational cost to generate a single token to $\mathcal{O}(n)$ from $\mathcal{O}(n^2)$ (where $n$ is the sentence length). Most existing sparse attention techniques ignore the specifics of this process and focus on sparsifying each attention operation separately. As non-attended tokens can still be attended to by subsequent tokens, memory benefits are limited. By contrast, our approach is compatible with this setting, allowing us to design an efficient batched data structure where dropped tokens are effectively removed from the computation.

## 3.1 Adaptively Sparse Attention

We allow the network to selectively drop parts of the context that are no longer required. An illustration of our proposed method can be seen in Fig. 2. At each layer, we introduce the parameters $\mathbf{W}^\ell_{Q_{\text{int}}}, \mathbf{W}^\ell_{K_{\text{int}}} \in \mathbb{R}^{d \times r}$ for dimension $r \in \mathbb{R}$, that calculate the interaction queries and keys $\mathbf{Q}^\ell_{\text{int}}, \mathbf{K}^\ell_{\text{int}} \in \mathbb{R}^{n \times r}$, as $\mathbf{Q}^\ell_{\text{int}} = \mathbf{X}^\ell \mathbf{W}^\ell_{Q_{\text{int}}}$ and $\mathbf{K}^\ell_{\text{int}} = \mathbf{X}^\ell \mathbf{W}^\ell_{K_{\text{int}}}$. We then calculate the *interaction* of token $k$ with token $j$ at layer $\ell$ as:

$$
\mathbf{I}^\ell_{k,j} = \begin{cases} \prod_{n=j+1}^k \bar{\mathbf{I}}^\ell_{n,j} \ \text{ and } \ \bar{\mathbf{I}}^\ell_{n,j} = \sigma\left( \frac{(\mathbf{Q}^\ell_{\text{int}})_n^\top (\mathbf{K}^\ell_{\text{int}})_j}{\sqrt{r}} + \beta^\ell \right), \text{if } j < k \\ 1, \text{if } j = k, \\ 0, \text{if } j > k, \end{cases} \tag{6}
$$

where $\sigma(\cdot)$ denotes the sparse sigmoid function introduced in Section 3.2 and $\beta^\ell \in \mathbb{R}$ a scalar parameter per layer, that controls the initial sparsity as seen in Fig. 3 (right). Indices in $\mathbf{Q}^\ell_{\text{int}}, \mathbf{K}^\ell_{\text{int}} \in \mathbb{R}^{n \times r}$ refer to the rows of the matrices. We can then modify the self-attention

$$
\text{SA}(\mathbf{Q}^\ell_i, \mathbf{K}^\ell_i, \mathbf{V}^\ell_i) = \text{softmax}\left( \frac{\mathbf{Q}^\ell_i (\mathbf{K}^\ell_i)^\top}{\sqrt{p}} + \log(\mathbf{I}^\ell) \right) \mathbf{V}^\ell_i. \tag{7}
$$

For $j > k$ we set $\mathbf{I}^\ell_{k,j} = 0$, which leads to masking entries in the self-attention, corresponding to the regular causal masking. We also impose that a token cannot drop itself, thus $\mathbf{I}^\ell_{k,k} = 1$. We want to preserve information regarding the current token as its predictions are particularly important in determining the next token for the regular language modeling task that we are considering. Small values of $\bar{\mathbf{I}}^\ell_{n,j}$ impose partial masking of the corresponding token in the attention, and complete masking occurs when $\bar{\mathbf{I}}^\ell_{n,j} = 0$. The cumulative product over tokens $j + 1 \to k$ in Eq. (6) imposes that dropping a token (when $\sigma(.) \to 0$) has an irreversible effect, as it will remain dropped for all subsequent tokens, and hence for the remaining of the generation process. The complexity of the pruning logic is $\mathcal{O}(n \cdot d \cdot r + n^2 \cdot r)$, which is lower than the one of the self-attention operation for $r < d$.

Our mechanism allows layers to act independently, meaning that different sparsity patterns are encountered across layers. We also experimented with tying the model's dropping decisions with depth by imposing that a token dropped at a given layer cannot be attended to in subsequent layers. However, we observed worse results and hence did not pursue this further. This is perhaps expected, given numerous results and interpretability studies regarding sparsity patterns of attention heads at different layers [Ramsauer et al., 2020, Hao et al., 2021].

## 3.2 Sparse Sigmoid

In Eq. (6), we use $\sigma(\cdot)$, as a sigmoid-like function to let the network decide when and what to drop. We favour binary decisions, leading to interaction values of either 0 or 1. Inspired by the $\alpha$-entmax function introduced in Peters et al. [2019], Martins et al. [2020], we define the $\alpha$-sigmoid (based on the entropies proposed by Tsallis [1988]) as:

$$
\sigma(x) = \alpha\text{-sigmoid}(x) = \text{argmax}_{p \in [0,1]} \left( p \cdot x + H_\alpha(p) \right), \tag{8}
$$

where

$$
H_\alpha(p) = \begin{cases} \frac{1}{\alpha(\alpha-1)} (p - p^\alpha + (1-p) - (1-p)^\alpha), \text{ if } \alpha \neq 1 \\ -p \log p - (1-p) \log(1-p), \text{ if } \alpha = 1. \end{cases} \tag{9}
$$

By varying $\alpha$ during the training, we can control the sparsity in the network, i.e. regulate the softness of the pruning mechanism. In practice, we start from small values of $\alpha = 1$ and increase it according to a cosine scheduler, as shown in Fig. 3. Small values of $\alpha$ allow meaningful gradient signals to pass through the dropping mechanism, which is crucial at the beginning of training. On the other hand, larger values of $\alpha$ lead to sparse results desired during inference. We thus increase $\alpha$ to values leading to very sparse solutions, as illustrated in Fig. 3. In practice, during inference, we replace $\sigma(\cdot)$ with the step function, that corresponds to $\alpha \to \infty$. We also initialize the biases parameters $\beta^\ell$ in (6) to a positive value, ensuring that tokens at the beginning of training have a prior towards not being dropped. This strategy also facilitates fine-tuning existing pretrained models, as our module

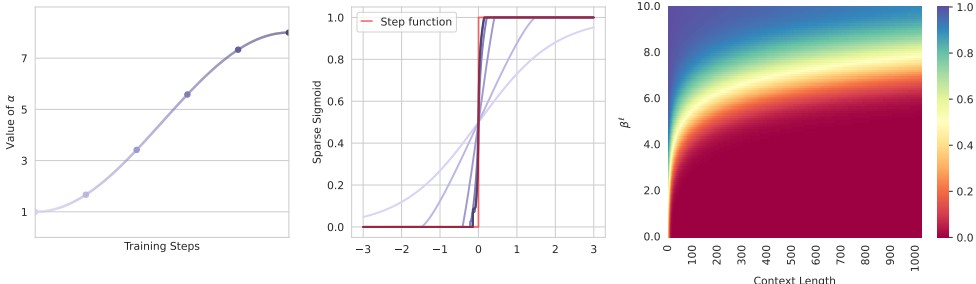

Figure 3: (Left) We use a cosine scheduler to set the values of $\alpha$ during training. (Middle) For values of $\alpha > 1$, mappings of the $\alpha$-sigmoid saturate at $\pm 1/(\alpha - 1)$. During inference, we replace the $\alpha$-sigmoid with a step function, that corresponds to the case $\alpha \to \infty$. (Right) Distribution of $\mathbf{I}_{k,j}^\ell$ for different values of $\beta^\ell$ with respect to the distance between the tokens $k - j$. For this depiction, we assume random normally distributed vectors as inputs and randomly initialized weights $\mathbf{W}_{Q_{\text{int}}}^\ell, \mathbf{W}_{K_{\text{int}}}^\ell$, according to 'He' initialization [He et al., 2015].

will initially default close to the identity function. The $\alpha$-sigmoid along with the training schedule on $\alpha$ allows for good signal propagation properties for the gradients [Noci et al., 2022]. We also explored using a regular sigmoid with a varying temperature [Kim et al., 2022], leading to suboptimal nonbinary predictions and instabilities during training. Training with our sparse sigmoid also directly eliminates the need to have any auxiliary network [Lee et al., 2023].

### 3.3 Regularized Objective

We augment the regular language modeling objective with a regularization that incentivizes the network $f$ to drop parts of the sequence. We fine-tune pretrained models, with parameters $\theta$, using the objective:

$$L(\theta, \mathbf{T}) = L_{lm}(\theta, \mathbf{T}) + L_{sparsity}(\theta, \mathbf{T}), \tag{10}$$

where

$$L_{lm}(\theta, \mathbf{T}) = \text{CE}(f_\theta(\mathbf{T}), \text{shift}(\mathbf{T})) \tag{11}$$

is the regular cross-entropy loss for the language modeling task based on the original and shifted input tokens $\mathbf{T}$, and

$$L_{sparsity}(\theta, \mathbf{T}) = \gamma \frac{2}{L\,n(n-1)} \sum_{i,\ell} \mathbf{I}_{i,j}^\ell \tag{12}$$

is the sparsity loss, encouraging the model to prune the context. In total $(L\,n(n-1))/2$ entries of $\mathbf{I}_{i,j}^\ell$ are learned, as indicated in Eq. (6). We choose $\gamma > 0$ to enforce different levels of sparsity. In general, for a current position $i$ in the context, we define as sparsity, the percentage of the previous tokens dropped, i.e. (tokens $\leq i$ dropped)$/i$.

## 4 Experiments

We fine-tune pretrained GPT-2 models [1], that support a context size of up to 1024 tokens, on a subset of the English Wikipedia *20220301.en* and English *bookcorpus* datasets. We keep a separate test set where we report perplexity after training. All models shown, for a fair comparison, were fine-tuned using the same lightweight training setup as described in Appendix A. When using our adaptive sparse attention, we use a cosine scheduler for the $\alpha$ parameter as displayed in Fig. 3 and specify $r = 64$ for the dimensions of $\mathbf{W}_{Q_{\text{int}}}^\ell, \mathbf{W}_{K_{\text{int}}}^\ell$. More ablations regarding optimization and variations of our dropping mechanism are provided in Appendix B. Unless otherwise stated, results refer to GPT-2-*small* models. We use the term *dense* for the regular GPT-2 models, fine-tuned without any additional $\mathbf{W}_{Q_{\text{int}}}, \mathbf{W}_{K_{\text{int}}}$ parameters.

---

[1]We use the pretrained models and tokenizers from https://huggingface.co/, for the GPT-2-{small, medium, large, xl} models. Here $n_{\text{vocab}} = 50257$.

**Baselines.** We compare against the baselines presented in Fig. 1. *Local Attention* refers to a causal attention mask, where each token attends to the previous $k$ tokens in the sequence, including itself. This can also be interpreted as restricting the receptive field of the model. *Sparse Attention* refers to the baselines from Child et al. [2019], Lin et al. [2022], where each token $i$ attends to the tokens satisfying (1) $\lfloor i/k \rfloor = \lfloor j/k \rfloor$ and (2) the tokens $k - 1, 2 \cdot k - 1, \ldots, \lfloor i/k \rfloor \cdot k - 1$ (numbering starts from zero). We fine-tune these baselines using the same aforementioned fine-tuning procedure, for different choices of $k$, leading to different levels of sparsity, depending on the current context size.

**Data structure.** Real-world deployment of our approach exhibits numerous challenges due to the nature of batched generation. In particular, we highlight differences in prompt length (initial prefix), different final lengths (termination criteria), and uneven dropping of tokens across different sentences. Maximum performance is achieved when the key-value cache is represented as a contiguous block of memory, and any masking resulting from padding or removed tokens ("holes") will result in a decrease in efficiency. To this end, we devise an efficient batched data structure that allows for efficient insertion and deletion of tokens (leveraging the set nature of the self-attention operation), while *(i)* allowing the underlying storage to be processed as a contiguous block of memory and *(ii)* ensuring that the load factor of the data structure is high enough to guarantee a performance speed-up. More details are provided in the Appendix A.

### 4.1 Results

**Perplexity vs sparsity.** We first study how context-pruning changes for different levels of sparsity in Fig. 4. Depending on the current context size, our method allows for up to 80% of the context to be successfully pruned, i.e. removed, with no performance loss in terms of perplexity (-0.085 average gain in perplexity when context size is 1000 tokens for 80.35% of sparsity compared to the dense counterpart). Our method also adapts to the current context size, meaning a network trained with specific sparsity regularization exhibits different levels of sparsity depending on the current context size. Compared to the baselines, our method exhibits consistently lower perplexity results for the same level of sparsity.

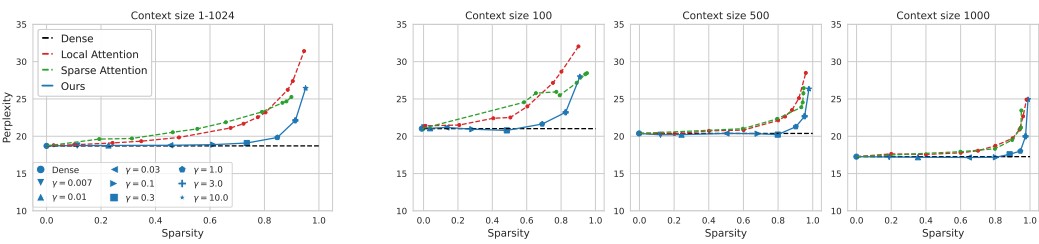

Figure 4: Perplexity (lower is better) for different levels of sparsity. (Left) Overall perplexity averaged across tokens with context size varying from 1 to 1024. The three plots on the right show perplexity for different context sizes.

**Zero-Shot Performance.** To test general model capabilities and complement perplexity evaluations, we provide results on several zero-shot tasks [Dettmers et al., 2022] in Fig. 5. Similar trends hold overall; our approach retains or even outperforms the performance of the dense baseline, even for cases with high sparsity. These tasks involve scenarios where the model is required to perform without any specific training or prior exposure to the target domain. The results obtained validate that the models' general capabilities can be retained, even under high levels of sparsity.

**Computational Analysis.** We analyse the gains in terms of FLOPs and required memory when generating new sequences due to caching in Fig. 6. Our dropping mechanism introduces additional computational costs for the calculation of $\mathbf{Q}_{\text{int}}^{\ell}, \mathbf{K}_{\text{int}}^{\ell}$ and the logic behind dropping via Eq. (6). Due to the relatively small chosen parameter $r$, i.e. the output dimension of the interaction weights $\mathbf{W}_{Q_{\text{int}}}^{\ell}, \mathbf{W}_{K_{\text{int}}}^{\ell}$, these are nevertheless minimal. Although the raw FLOPs benefit when using sparse models does not seem very significant, as aforementioned, inference is predominately memory-bound. The attention thus takes a significant proportion of real-time inference [Dao et al., 2022]. On the contrary, dense matrix multiplications used for all linear projections are very efficient. Memory

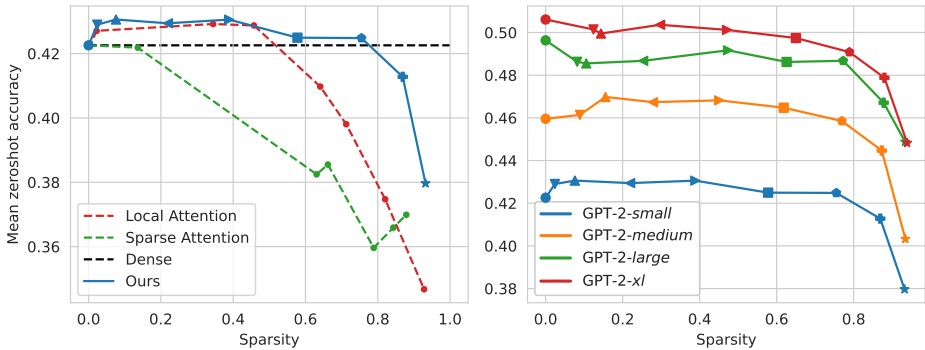

Figure 5: Mean zero-shot accuracy (higher is better) for the *WinoGrande*, *HellaSwag*, *PIQA*, and *LAMBADA* datasets. As the sparsity of all methods depends on the context size, we average the expected sparsity based on the lengths of the prefixes in these datasets. (Left) GPT-2-*small* models and (right) all GPT-2 models.

benefits, on the other hand, are substantial, as the memory required for caching is a linear function with respect to sparsity, with a negative slope. Sparser solutions will thus additionally allow us to generate more sequences in a batched fashion. This is particularly relevant for bigger models, also longer sequences, where batch decoding is a major challenge [Shazeer, 2019].

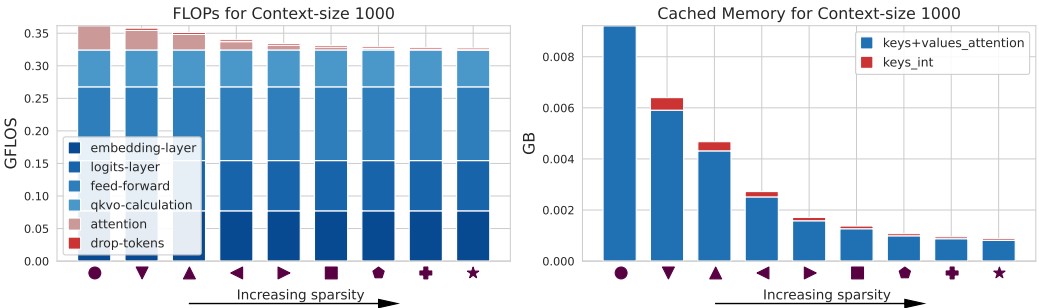

Figure 6: (Left) Distribution of FLOPs for models with different levels of sparsity. Here, *embedding-layer* refers to the embedding of the input sequence to the representation $\mathbf{X}^0$, *logits-layer* to the projections of the final representation $\mathbf{X}^L$ according to the vocabulary size, *feed-forward* to the feed-forward components, summed across the different layers, *qkvo-caclulation* to the projection of the current representation to queries, keys, values and the final output projection, *attention* to the actual softmax operation and *drop-tokens* to additional compute required for calculating $\mathbf{Q}_{\text{int}}^\ell, \mathbf{K}_{\text{int}}^\ell$ and performing dropping via Eq. (6). (Right) Memory requirements when caching previous activations (keys and values). When implementing dropping, interaction keys $\mathbf{K}_{\text{int}}^\ell$ have to be additionally cached.

**Throughput.**    We demonstrate how reduced context and reduced memory requirements can lead to significant real-world time throughput in Fig. 7. Initially, our pruned networks are slower in terms of latency for small context lengths, because of the additional cost associated with the logic behind pruning. Nevertheless, they quickly surpass the dense baseline that struggles as the context size increases. This verifies the fact that although raw FLOPs benefits look unsubstantial, in fact, this leads to significant gains due to the specific memory profile of Transformers' inference. Crucially, our pruned networks can support a much bigger batch size, leading to significant throughput gains. More specifically, for long context sizes, our GPT-2-*small* model offers an additional $98\%$ margin in throughput for a loss in perplexity of only $0.316$, with respect to the dense counterpart. Similarly, our GPT-2-*medium* model can yield $189\%$ additional throughput for only $0.084$ loss in perplexity for a context size of 1000 tokens. In particular, the same model (for $\gamma = 1.0$) provides a higher throughput than a GPT-2-*small* model, while achieving $3.769$ lower perplexity. As context windows become larger by the day in state-of-the-art models, we expect these gains to become even more relevant.

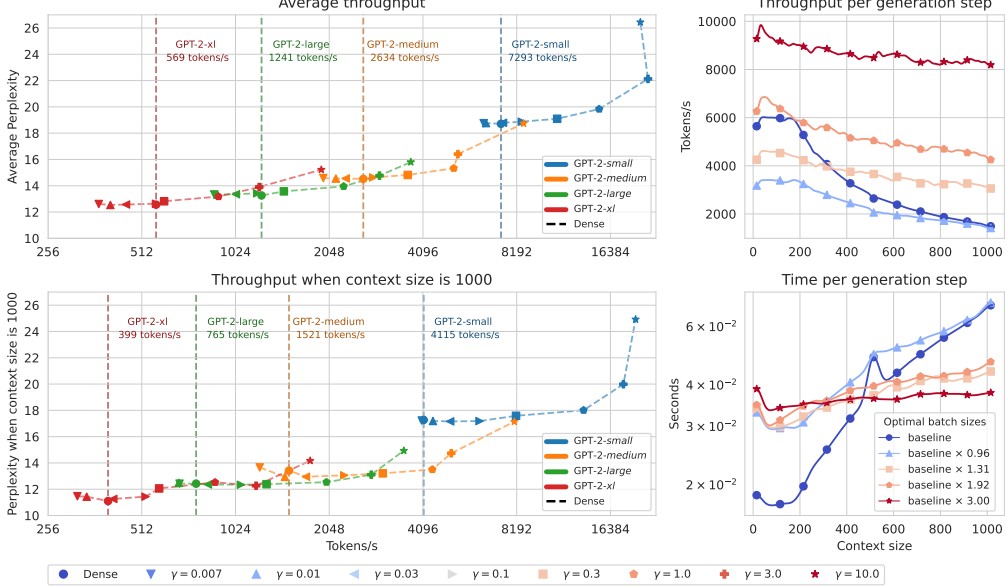

Figure 7: We measure throughput using the optimal batch size on an NVIDIA RTX A5000 GPU. (Left) Throughput in terms of tokens per second for different models and different levels of sparsity (top) averaged across tokens for context sizes from 1 to 1024 and (bottom) when the context size is 1000 tokens. (Right) Average (top) throughput for varying context size for the GPT-2-*medium* model and average (bottom) time per generation step for varying context size. As our models require significantly less memory, a larger batch size can be accommodated, where large portions of the throughput gains can be attributed to.

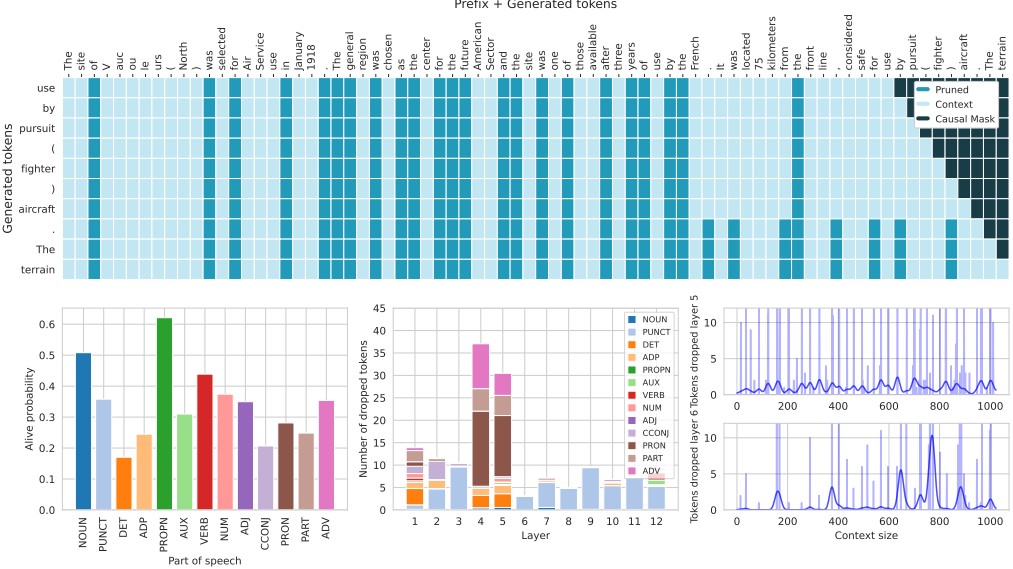

Figure 8: (Top) Example of pruned tokens for layer 5 for the GPT-2-*small* model fine-tuned with $\gamma - 0.3$ during generation. Most pruning is triggered by punctuation. (Bottom-left) We calculate the probability of tokens to be kept in the context based on the part of speech (POS) of the words they correspond to. (Bottom-middle) Most dropping is caused by tokens corresponding to punctuation, but distinct layers behave differently. (Bottom-right) Example of the number of tokens pruned by the tokens' position id, for 2 layers of GPT-2-*small*.

**Interpretability.** Fig. 8 provides insights into the interpretability aspect of the model's decision-making process. It is observed that token removal predominantly occurs when encountering stop words (punctuation), which aligns with the intuition that local information within a sentence becomes less relevant after its completion. Furthermore, it is worth noting that layers at varying depths exhibit distinct behaviours, reinforcing our rationale for dissecting token removal decisions across depth. The variance in sparsity distribution across different depths indicates the necessity of conducting additional interpretability research to obtain valuable insights in the interactions of the tokens within the model. We provide more insights towards this direction in the Appendix C.

## 5   Discussion

We proposed Adaptively Sparse Attention, a novel approach to dynamically prune the context in decoder-only Transformer architectures. Our results indicate that our technique performs favourably compared to competitive baselines in terms of the ratio between perplexity and sparsity of the attention weights. Remarkably our approach also significantly reduces the computational and memory requirements without affecting its final performance. We practically showcase these benefits achieving more than double the throughput at cases. Adaptively sparse attention comes with two additional practical advantages: first, it can be seamlessly integrated into existing pre-trained models via a cheap fine-tuning step; second, it represents an orthogonal contribution to the burgeoning research line aimed at increasing the level of efficiency of modern LLMs. As such, we envision its combination with existing techniques like weight pruning and quantization to be a promising avenue for future research.

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

# A  Experimental Setup

We use the pretrained GPT-2 models from huggingface. Parameters of the architecture for these models are provided in Table 1.

| Name | Parameters | Number of layers | Number of heads | Model Dimension $d$ |
|---|---|---|---|---|
| GPT-2-*small* | 124M | 12 | 12 | 768 |
| GPT-2-*medium* | 350M | 24 | 16 | 1024 |
| GPT-2-*large* | 774M | 36 | 20 | 1280 |
| GPT-2-*xl* | 1558M | 48 | 25 | 1600 |

Table 1: Parameters of the architecture for the GPT-2 models.

**Training.**  We fine-tune pretrained models on a subset of the English Wikipedia *20220301.en* and English *bookcorpus* datasets, for a total of 25000 steps with a batch size of 6. The datasets are provided by huggingface at https://huggingface.co/datasets/wikipedia and https://huggingface.co/datasets/bookcorpus respectively. We use a learning rate of $1e^{-4}$ for the *small* and *medium* models and $5e^{-5}$ for the *large* and *xl* models with the Adam optimizer. We do not use any weight decay or any scheduler for the learning rate. For the self-attention operations we use *flash-attention* as provided by the *scaled_dot_product_attention* in *pytorch-2.0*[2].

**Evaluation.**  For the zero-shot accuracy experiments, we report accuracy on the WinoGrande [Sakaguchi et al., 2021], HellaSwag [Zellers et al., 2019], PIQA [Bisk et al., 2020] and LAMBADA [Paperno et al., 2016] datasets, similar to Dettmers et al. [2022]. As samples in these datasets have different lengths, we report as sparsity the mean sparsity over the samples for which predictions were made.

**Efficient Memory Allocation.**  As we explained in Section 4, the computational cost of our method is greatly affected by the underlying data structure used for representing the key-value cache. Conceptually, the data structure should implement the following methods (all batched):

- push(): inserts a new token ($\mathbf{K}$, $\mathbf{V}$, and $\mathbf{K}_{\text{int}}$).
- get(): returns the keys/values added so far as a contiguous block of memory, as well as a binary mask used to represent padding and potential gaps due to removed tokens.
- remove(): given a binary mask of the same shape as that returned by get(), removes the specified tokens from the data structure.

Ideally, the insertion and deletion operations should be as efficient as possible, while guaranteeing that the memory block returned by get() is as packed as possible (high load factor). Additionally, the data structure should support dynamic resizing as more tokens are inserted. A simple (yet inefficient) baseline consists in implementing the above interface as a *dynamic array* (i.e. a preallocated buffer that is dynamically resized once full) where erased tokens are simply masked out. Such an implementation, while correct, does not result in any memory and computation savings. Instead, motivated by the intuition that *self-attention* is a set operation – meaning that tokens do not need to be stored in sequential order – we recycle the memory slots of erased tokens. We insert new tokens at the leftmost available position in the data structure (Fig. 2), and ensure that the *load factor* (the ratio between the length of the longest sentence $n$ and the capacity of the buffer) is always greater than a specified value. We choose a load factor of $0.9$ and dynamically consolidate the data structure when the effective load factor falls below this threshold. We also mention that the asymptotic cost of these operations does not have a significant impact on the final performance, as the overall cost is still dominated by the $\mathcal{O}(n)$ cost of the self-attention mechanism (for a single generated token). In our implementation, both push() and remove() have a cost of $\mathcal{O}(n)$, while get() has a cost of $\mathcal{O}(1)$ since it simply returns a view of the memory buffer. We also experimented with asymptotically faster implementations for push() and remove() (at $\mathcal{O}(\log n)$ using a priority queue), but found these to be slower in practice due to inefficient use of the GPU.

---

[2]https://pytorch.org/docs/stable/generated/torch.nn.functional.scaled_dot_product_attention.html

# B Training Results and Ablations

**Training Curves.** We provide an example of a training curve in Fig. 9. Initially, the loss of the model is high, as at initialization many of the tokens are dropped, despite the introduced $\beta_\ell$ parameters. Higher values of $\beta^\ell$ can significantly mitigate this phenomenon. We noticed however that very high values of $\beta^\ell$ lead to worse final solutions in terms of achieved complexity for a given level of sparsity. For our experiments we initialize $\beta^\ell = 2.0, \forall \ell \in \{1, 2, \ldots, L\}$.

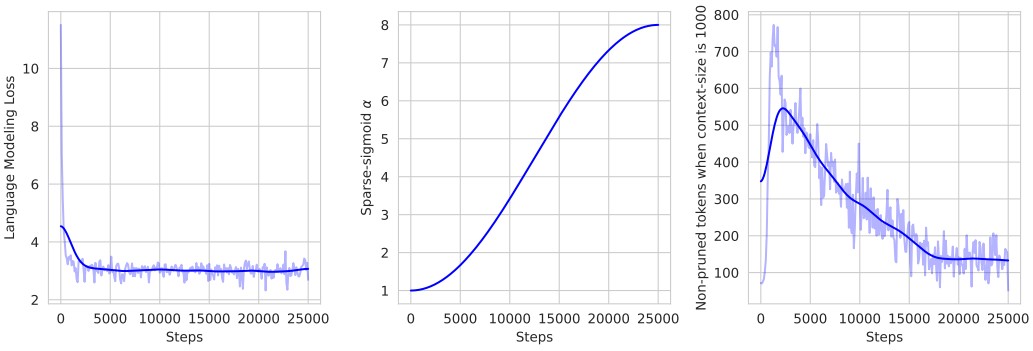

Figure 9: Training curve for the GPT-2-*small* model trained with a regularizer $\gamma = 0.3$.

**Setting $\alpha$.** As the value of $\alpha$ in the sparse-sigmoid functions rises, solutions become sparser, as indicated by the sparsity in Fig. 9 (right). During inference, we want to replace the sparse-sigmoid function with a simple step function, thus we should make the sparse sigmoid as similar as possible to the step function during training, to mitigate any inconsistencies caused by functional differences. In practice, we found no benefit from increasing $\alpha$ to values larger than $8.0$. The speed by which $\alpha$ is increased also plays a significant role. Increasing $\alpha$ too quickly does not allow for the new interaction parameters to be optimized correctly, leading to suboptimal solutions. We found that a cosine scheduler for 25000 steps was enough to lead to solutions of adequate sparsity. We present results when optimizing for a different number of steps in Fig. 10.

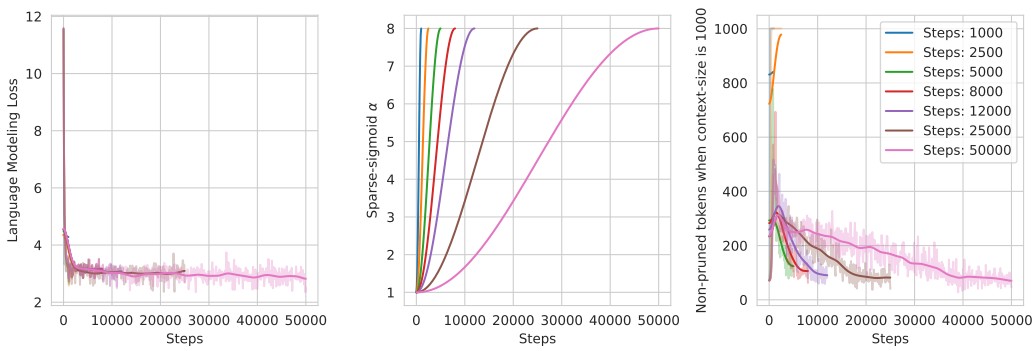

Figure 10: Training curves when a different number of total steps is executed, using the same cosine scheduler for the values of $\alpha$.

**Setting $r$.** For our experiments, we used $r = 64$ for the embedding dimensions of the interaction weights. We experimented with different dimensions in Fig. 11. Larger dimensions lead to higher sparsity for the same perplexity but also require more memory to store the interaction keys. We found $r = 64$ to be a good compromise and tradeoff between extra memory and performance for the same sparsity.

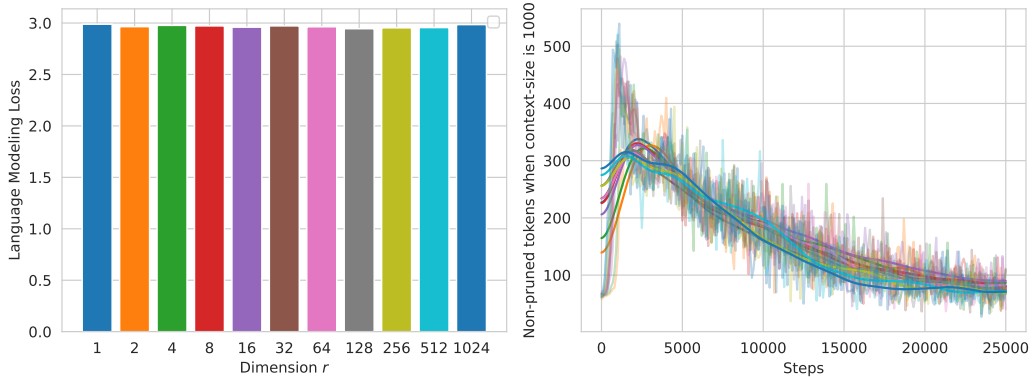

Figure 11: Training curves when varying the dimension $r$. (Left) Final language modelling loss and (right) sparsity during training. For different dimensions $r$, similar perplexity is achieved. Larger dimensions $r$ generally allow for sparser solutions for the same perplexity.

**Propagation of Pruning with Depth.**  We also experimented with tying the dropping decisions with depth. In this case we modify Eq. (6) as:

$$
\mathbf{I}_{k,j}^{\ell} = \begin{cases} \prod_{p=1}^{\ell} \prod_{n=j+1}^{k} \bar{\mathbf{I}}_{n,j}^{p} \ \text{ and } \ \bar{\mathbf{I}}_{n,j}^{p} = \sigma \left( \frac{(\mathbf{Q}_{\text{int}}^{p})_{n}^{\top} (\mathbf{K}_{\text{int}}^{\ell})_{j}}{\sqrt{r}} + \beta^{p} \right), \text{if } j < k \\ 1, \text{if } j = k, \\ 0, \text{if } j > k. \end{cases} \tag{13}
$$

Dropping a token in this case at a layer $\ell$ directly enforces that the token is dropped for all subsequent layers. Such a choice is inspired by similar pruning techniques in Transformer encoder models [Bolya et al., 2022] that usually, reduce the number of tokens with depth. We found, however, that this choice led to some disadvantages.

Firstly, we determined that sparsity is not a monotonic function of depth. Similar results have been shown before [Ramsauer et al., 2020]. Typically, the middle layers can be more sparse compared to the first and last layers. Secondly, for deeper models (GPT-2-*large* and GPT-2-*xl*), we found that the large number of elements over which the cumulative product is taken over in Eq. (13), led to difficulties in learning. This can be perhaps expected given that no such training objective was optimized during the pre-training phase of these models. Finally, propagating the pruning decisions across depth significantly complicates the challenges for an efficient implementation.

**Freezing Initial Parameters.**  We experimented with training just the interaction weights $\mathbf{W}_{Q_{\text{int}}}^{\ell}, \mathbf{W}_{K_{\text{int}}}^{\ell} \in \mathbb{R}^{d \times r}, \beta_{\ell} \in \mathbb{R}$, leaving the rest of the parameters of the network frozen. This led to an average increase in the perplexity of 9.285 for the same levels of sparsity, showing that modifications to the network's weights/logic are still necessary.

## C   Additional Results

**Sparsity per Layer.**  In Fig. 12 we present the average level of sparsity per layer. In Fig. 13 we present similarly the number of tokens kept per layer for different sizes of the initial context.

**Tokens that Cause Pruning.**  In Fig. 8 we presented which kind of tokens cause the pruning to take place. For the same example and settings, we present here exactly which tokens are dropped and when, in Fig. 14.

**Context Switch.**  To better understand how well the dropping mechanism works, we create an artificial example, where we concatenate together text from three distinct, independent contexts. More specifically, we concatenate together the texts, as seen in Table 2. Inspecting the attention matrices in Fig. 15, reveals that the network has largely discovered the switches in the context and learned to ignore most of the preceding text if that comes from a different context.

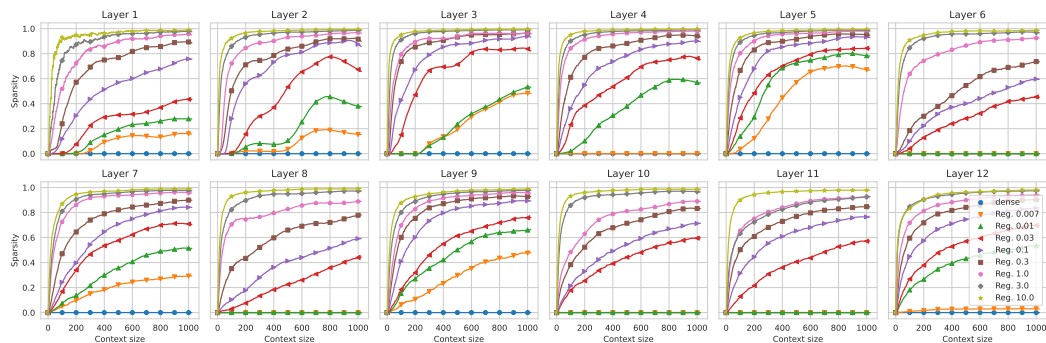

Figure 12: Sparsity per layer for different levels of regularization $\gamma$. Here we are averaging predictions for different context sizes ranging from 1 to 1024.

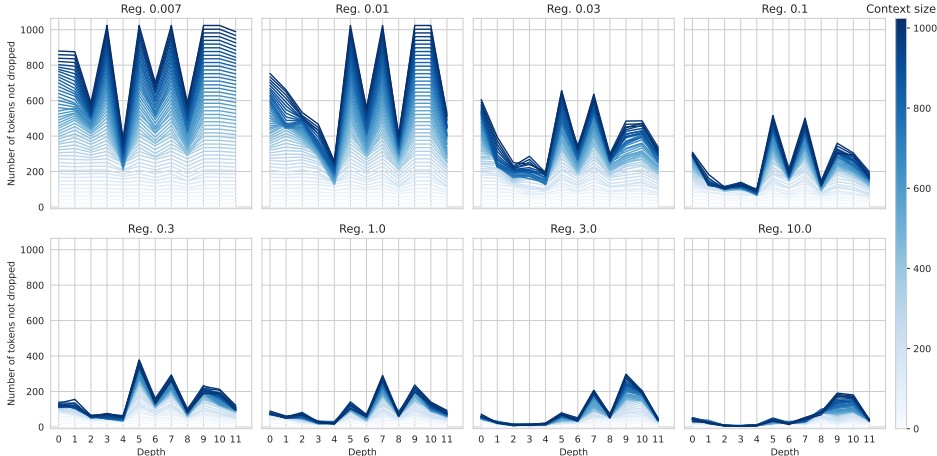

Figure 13: Sparsity per layer for different levels of regularization $\gamma$. Different colors indicate different initial un-pruned context sizes.

| |
| --- |
| Lebanon, officially the Republic of Lebanon or the Lebanese Republic, is a country in Western Asia. It is located between Syria to the north and east and Israel to the south, while Cyprus lies to its west across the Mediterranean Sea; its location at the crossroads of the Mediterranean Basin and the Arabian hinterland has contributed to its rich history and shaped a cultural identity of religious diversity. |
| A motorcycle (motorbike, bike, or trike (if three-wheeled)) is a two or three-wheeled motor vehicle steered by a handlebar from a saddle-style seat. Motorcycle design varies greatly to suit a range of different purposes: long-distance travel, commuting, cruising, sport (including racing), and off-road riding. Motorcycling is riding a motorcycle and being involved in other related social activity such as joining a motorcycle club and attending motorcycle rallies. |
| Zorba's Dance is an instrumental by Greek composer Mikis Theodorakis. The song featured for the dance, which has become known as sirtaki, in the 1964 film Zorba the Greek, for which Theodorakis wrote the soundtrack, and became renowned around the world. It is now commonly played and danced to in Greek tavernas. The film's track has since been recorded as a standalone song by many different musicians from around the world. |

Table 2: Concatenated contexts used for the context switch example in Fig. 15.

**Ideal Generation Speed-up.** In Fig. 7 we presented throughput in a realistic scenario where 'holes' formed in the memory buffer in an uneven fashion between samples in the batch. We expect that as we employ our method for larger models and contexts, the maximum feasible batch size will be reduced, mitigating partly this unwanted phenomenon. To evaluate the maximum possible speed-up,

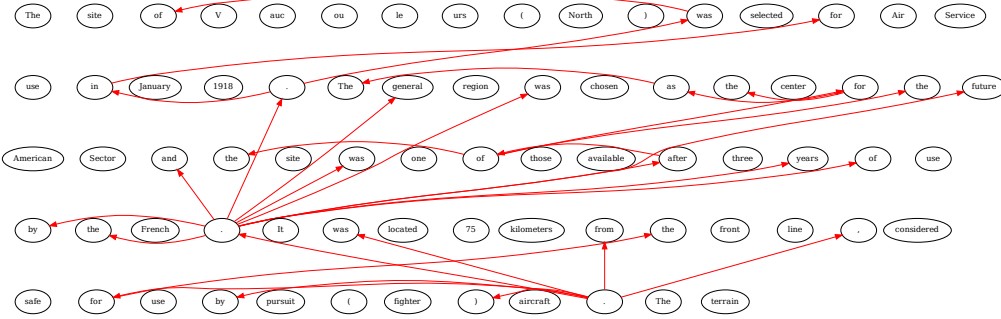

Figure 14: We illustrate during generation which tokens cause pruning. The same layer and model is used as in Fig. 8. Arrows indicate decisions to prune. Nodes correspond to tokens, as determined by the tokenizer.

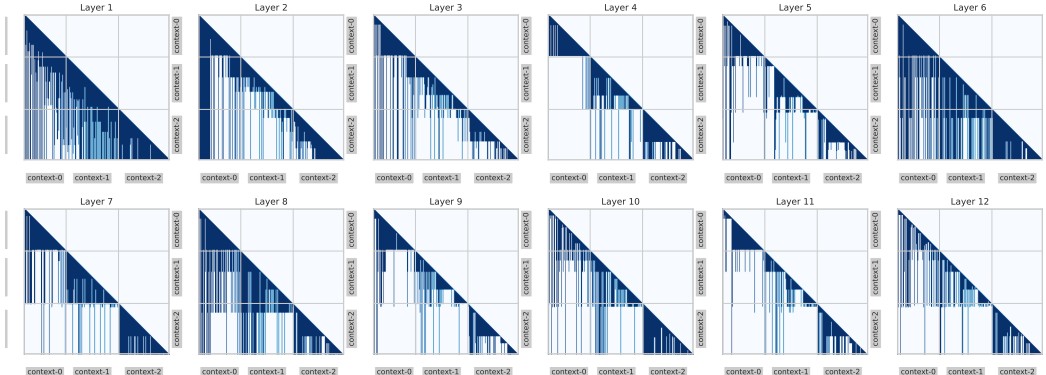

Figure 15: Attention weight for different layers for the context switch example in Table 2. Color here indicates that the token can be attended to and does not correspond to the actual attention weight. Notice the casual masking and the three emerging dense triangular sub-matrices, especially in layers 7, 8, 9 and 10.

we generate samples using the same prefix and the same sampling strategy across samples. The holes generated in this case are the same across the batch. We demonstrate the throughput achieved in this case in Fig. 16. The benefits in this case are a lot more clear.

**Investigating Dropped Tokens.**   In Transformer encoder models, it has been shown that similar tokens can be successfully pruned in deeper layers (e.g. Bolya et al. [2022]). Similarity, in this case, is measured by calculating the cosine similarity of the keys $\mathbf{K}$ across different tokens. To test how feasible such a pruning strategy is in our scenario, we calculate for the keys $\mathbf{K}$ of each token, the minimum cosine distance to other tokens in the sequence. We then group these distances based on whether the token was subsequently dropped or not, in Fig. 17. We can see that tokens that have other tokens very similar to them in the sequence are pruned. However, tokens with no other similar ones in the sequences can also be dropped.

We experimented with the framework from Bolya et al. [2022], which has exhibited very successful results for transformer encoder models out of the box, i.e. without any additional fine-tuning. We found, nonetheless, that the perplexity quickly diverged, i.e. increased, even for small levels of sparsity. This constitutes another indication that pruning decoder models requires additional effort. Compared to encoder models, decoders make a different prediction for each of the tokens in the input, and token similarity by itself is not good evidence for pruning. We finally highlight that even if the method from Bolya et al. [2022] achieved comparable results, computational benefits will be

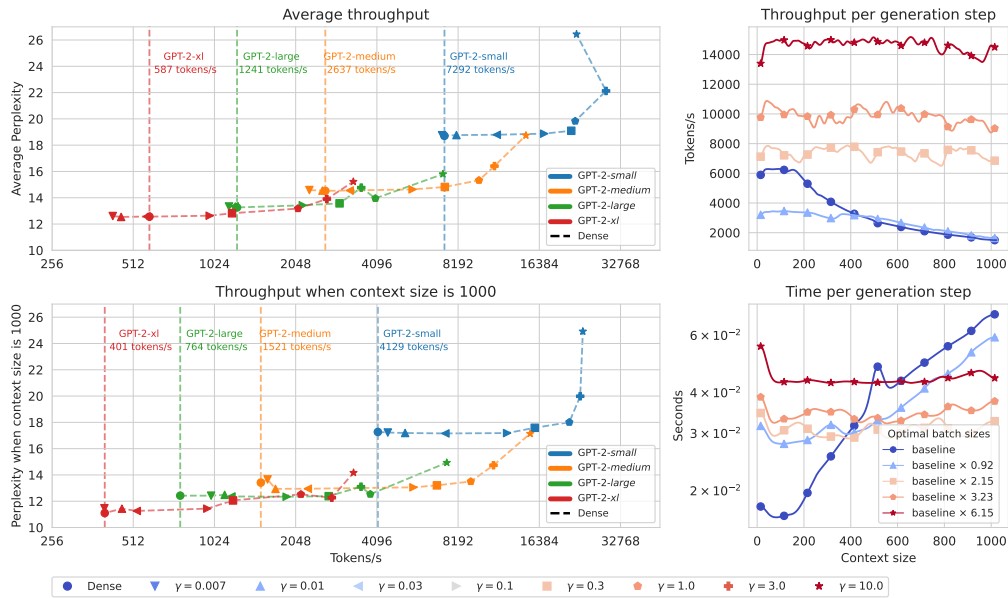

Figure 16: Maximum potential speed-up from our method achieved by the homogeneity of the memory allocation.

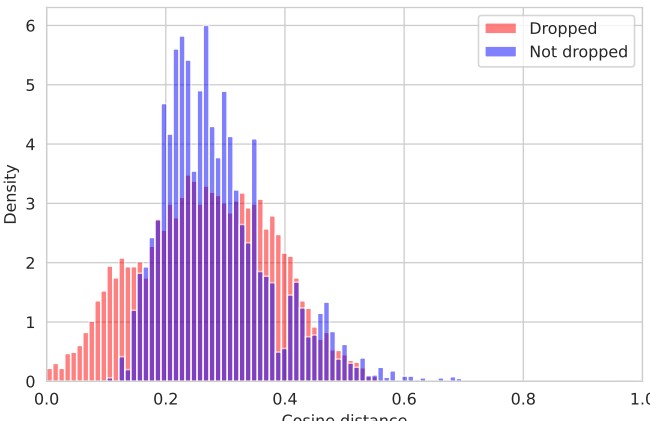

Figure 17: Minimum cosine similarity to other non-pruned tokens in the sequence. We group the tokens, based on whether these were subsequentially dropped or not. Results are averaged across samples and layers.

negligible, as no memory benefits can be achieved in this case and generation is performed one token at a time.

**Attended Tokens across Layers.** We provide additional visualization of the attended tokens in Fig. 18. Different layers exhibit different levels of sparsity and attend to different tokens.

**How sparsity affects specific NLP tasks** Evaluating GPT models pretrained by language modelling objective is in general challenging and different techniques have been developed based on the final intended use. This challenge motivated us to showcase zeroshot accuracy to downstream tasks, in addition to evaluating perplexity for upstream tasks. To better check if some tasks are affected more by sparsity compared to others, we provide per-task zeroshot performance, see Figure 19. We

Figure 18: We visualize attended tokens across layers. Color indicates the ability to attend to a token and not the actual attention weight.

also include zeroshot results for additional tasks, that require long-range dependencies. It becomes clear that some tasks are affected more than others, according to our intuition.

**Inference Compute** An additional benefit of our pruning strategy is that we can accommodate longer initial contexts for the same hardware. Table 3 showcases the maximum potential context windows for different batch sizes. Our strategy accommodates inference with much longer contexts, for a fixed device.

**Additional Models** As the original context grows, pruning the context makes an increasingly significant difference, as more opportunities for pruning exist. Generally, enlarging the context length of Transformers is a fascinating area of research with novel techniques being currently proposed. In our experiments, we are limited by the maximum positional encodings of the base GPT models. To evaluate how our method performs in larger contexts, we additionally present preliminary results

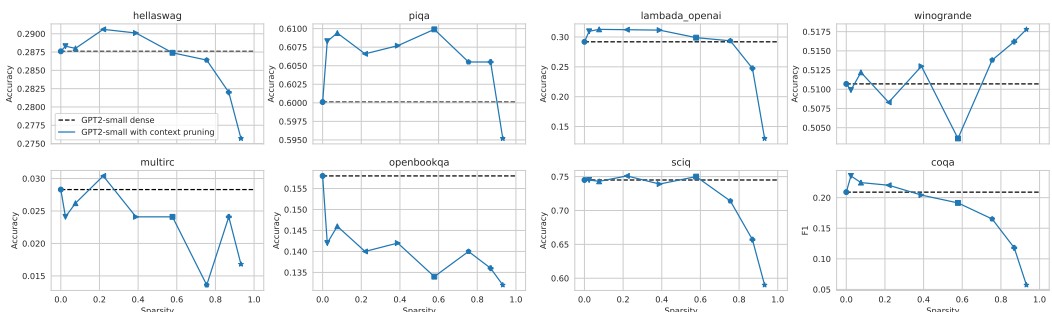

Figure 19: Zeroshot performance for individual tasks and different sparsity levels. Different tasks capture different capabilities of the model and may be affected differently by the varying enforced sparsity in the context. *Hellaswag* is a challenge dataset for evaluating commonsense natural language inference, *piqa* evaluates physical commonsense knowledge, *lambda_openai* is a reasoning challenge that evaluates the capabilities of computational models for text understanding by means of a word prediction task, *winogrande* is a benchmark for commonsense reasoning. Complementary, we also present results for *multirc* which is a dataset of short paragraphs and multi-sentence questions, *openbookqa* that is a multiple-choice elementary-level science questions, *sciq* evaluates science exam questions and *coqa* measures the ability of machines to understand a text passage and answer a series of interconnected questions that appear in a conversation. We believe some of these datasets better capture longer contexts and longer-range interactions.

| BATCH-SIZE \ MODEL | BASELINSE | 50% SPARSITY | 70% SPARSITY | 90% SPARSITY |
|---|---|---|---|---|
| 16 | 1729 | 3312 | 5434 | 16221 |
| 32 | 865 | 1616 | 2653 | 7901 |
| 64 | 433 | 784 | 1274 | 3742 |

Table 3: Maximum possible sequence length (number of tokens) for different batch sizes and different levels of sparsity for the *GPT-2-xl* models, on a single NVIDIA RTX A5000. *X% Sparsity* denotes our model that uses adaptively sparse attention.

on pythia language models. Pythia uses rotary position embedding, as many current state-of-the-art LLMs. We present results on 'Perplexity vs sparsity' in Figure 20. These concrete results demonstrate that our pruning technique *works out of the box* for different positional encodings as well. Longer contexts supported by our base model (for pythia models that is 2048) additionally allow for higher levels of sparsity for the same performance. In the future, we intend to also evaluate larger models, potentially with the use of LoRA [Hu et al., 2021]. We again highlight that our approach is not specific to a particular language model, as it can be applied to any autoregressive transformer architecture.

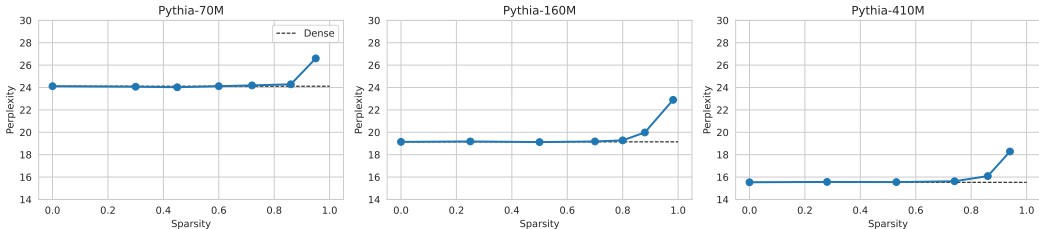

Figure 20: Perplexity (lower is better) for different levels of sparsity. Perplexity is averaged across tokens with context size varying from 1 to 2048. Three subplots correspond to pruning three different base *pythia* models.

# D    Discussion

We presented a simple approach that can be applied to any decoder-based Transformer autoregressive model. In an era where big foundation models are taking the community by storm, we presented a relatively inexpensive technique that can be applied to these models and can lead to significant gains in terms of computational resources required to perform inference. We truly believe the balance between model performance and resource efficiency is crucial for the widespread adoption of large-scale language models, and our approach offers a promising avenue for achieving this goal. We hope that our technique offers more interpretable predictions and inspires future research.

**Reproducibility.**    We have taken multiple steps to ensure the reproducibility of our experiments. We refer the reader to Section 4 and Appendix A for a complete description of the training protocol. We have also released the code as part of the supplementary material, including scripts on how to reproduce our results. Additionally, trained models will be released for further research.

**Limitations.**    Our work focuses on autoregressive generation models based on Transformers. Although we argue that the same technique can be applied out-of-the-box to any such model, we focused on text generation models and specifically the GPT-2 family. Due to the uniformity of recent architectures, we expect that these results generalize to other popular models.

