# OpenReview forum: "Dynamic Context Pruning for Efficient and Interpretable Autoregressive Transformers"
_NeurIPS.cc/2023/Conference — NeurIPS 2023 spotlight_

### Official Review · Reviewer_jYJo · 2023-07-03

**Soundness:** 3 good
**Presentation:** 4 excellent
**Contribution:** 3 good
**Rating:** 7
**Confidence:** 3

**Summary:**

This paper proposes an efficient and interpretable context-based dynamic pruning method. They use additional query / key layers to generate dynamic attention masks and sparsify self-attention maps. They also introduce a sparse sigmoid and regularization term to control the sparsity. In experiments, they demonstrate the lowest performance degradation compared to previous static attention pruning methods at the same level of sparsity. The results of throughput and speed analysis show that the proposed method can achieve additional inference efficiency with minimal performance loss. They analyze the distribution of remaining contexts with respect to part of speech and the depth of the layer. They also demonstrate that the proposed method dynamically prune attention based on contexts through context switch experiments.

**Strengths:**

- The proposed method is well-motivated and easy to follow.
- The proposed method efficiently increases interpretability and sparsity with minimal performance drop compared to previous research.
- They provide an extensive analysis of the relationship between context length, throughput, and speed in various parameter sizes.
- Based on the proposed pruning method, they devise efficient batched data structures for the optimized computation.

**Weaknesses:**

- Quantitative comparison of throughput and speed with local/sparse attention would contribute to a comprehensive understanding.
- Further qualitative study of interpretability (Fig. 8) varying the sparsity level (gamma) would provide additional intuitive observations.

**Questions:**

Do you have expectations on how the zero-shot performance difference between the pruning strategies will change as the number of model parameters increases?

**Limitations:**

As written in the Limitations section, scalability studies on larger language models (>7B) would provide further insights into the dynamic pruning method.

---

> ### Author Rebuttal · Authors · 2023-08-08
>
> We thank the reviewer for the feedback and the interesting questions. We are glad that the reviewer found the method easy to follow, and we hope our work sparks future work in the direction of efficient and more interpretable inference. In the following, we address comments made.
>
> > Quantitative comparison of throughput and speed
>
> Local and Sparse attention have a fixed logic in determining which tokens to drop. These methods then, require fewer FLOPs for the same level of sparsity. For completeness, we present here preliminary results on the throughput for different levels of sparsity for a GPT-2-xl model on an NVIDIA RTX A5000.
>
> ```
> +------------------------------------------------------+
> |Sparsity \ Model|Local Attention|Sparse Attention|Ours|
> +----------------+---------------+----------------+----+
> |       0.4      |      640      |       641      | 570|
> +----------------+---------------+----------------+----+
> |       0.6      |      850      |       851      | 765|
> +----------------+---------------+----------------+----+
> |       0.8      |      1391     |      1390      |1190|
> +------------------------------------------------------+
> ```
> Numbers indicate throughput in tokens/s, when the original unpruned context length is 1000 tokens. Although Local and Sparse Attention mechanisms allow for slightly higher throughput, these are counteracted by a higher drop in performance. This is captured both when measuring upstream perplexity (Figure 4), and especially when evaluating zeroshot performance (Figure 5).
>
> > Further qualitative study of interpretability.
>
> Interpreting attention weights is in general a challenging task. Our sparse attention is in that regard unique for two reasons. Firstly, only a subset of the context tokens are being attended to, so attention weights are concentrated only on the more important parts of the context. This allows us to study the significance of each token, depending on if it is being dropped or not. Secondly, by analyzing which tokens trigger pruning, we can see better understand at what time and which parts of the context become irrelevant. We have included more results in Appendix C, which we hope might help towards grasping a better intuition. Specifically, we present the sparsity patterns per layer in Figures 12 and 13. We also visualize the mechanism of dropping in Figure 14, including in an artificial scenario where we expect dropping to occur, by concatenating paragraphs of different content, in Figure 15. We also provide some additional results on the embedding nature of the dropped tokens.
>
> However, we agree with the reviewer that more possibilities exist to study and interpret the nature of this attention. We hope that our approach offers concrete advantages towards such an interpretation. Some related references [1, 2, 3, 4].
>
> Preliminary and complementary to Figure 8 (bottom-left) we present the alive probability for different parts of speech elements and models with different sparsity patterns. As $\gamma$ increases, the alive probability of different parts of speech decays differently.
>
> ```
> +---------------------------------------+
> |gamma| NOUN |  DET |PROPN| VERB |  ADV |
> +-----+------+------+-----+------+------+
> | 0.0 | 0.967| 0.940|0.985| 0.970| 0.982|
> +-----+------+------+-----+------+------+
> | 0.0 | 0.919| 0.763|0.964| 0.935| 0.932|
> +-----+------+------+-----+------+------+
> | 0.0 | 0.850| 0.532|0.884| 0.822| 0.783|
> +-----+------+------+-----+------+------+
> | 0.1 | 0.672| 0.305|0.775| 0.648| 0.587|
> +-----+------+------+-----+------+------+
> | 0.3 | 0.508| 0.170|0.621| 0.439| 0.354|
> +-----+------+------+-----+------+------+
> | 1.0 | 0.309|0.0851|0.439| 0.253| 0.212|
> +-----+------+------+-----+------+------+
> | 3.0 | 0.159|0.0506|0.274| 0.127| 0.135|
> +-----+------+------+-----+------+------+
> | 10. |0.0803|0.0338|0.132|0.0455|0.0553|
> +---------------------------------------+
> ```
>
> [1] Chefer, Hila, Shir Gur, and Lior Wolf. "Transformer interpretability beyond attention visualization." Proceedings of the IEEE/CVF conference on computer vision and pattern recognition. 2021.
>
> [2] Bolya, Daniel, et al. "Token merging: Your vit but faster." arXiv preprint arXiv:2210.09461 (2022).
>
> [3] Rigotti, Mattia, et al. "Attention-based interpretability with concept transformers." International Conference on Learning Representations. 2021.
>
> > How will zero-shot performance change as the number of model parameters increases?
>
> In our conducted experiments involving models with varying numbers of parameters, a consistent trend has emerged. Specifically, we have consistently observed that models employing pruned contextual components exhibit the capability to retain the performance levels of their unpruned counterparts, even when subjected to notably high sparsity levels.
>
> The continuous evolution and advancement of efficient fine-tuning approaches [4, 5] gives us confidence that the extensibility of these techniques to even larger models (>7B) is within reach, despite computational constraints. Looking ahead, we are particularly enthusiastic, as this avenue of investigation holds substantial promise, particularly given its potential for effective deployment within real-world systems.
>
> [4] Hu, Edward J., et al. "Lora: Low-rank adaptation of large language models." arXiv preprint arXiv:2106.09685 (2021).
>
> [5] Dettmers, Tim, et al. "Qlora: Efficient finetuning of quantized llms." arXiv preprint arXiv:2305.14314 (2023).
>
> We thank the reviewer for the interesting points raised, which will be discussed in the main text.

---

> > ### Comment · Reviewer_jYJo · 2023-08-16
> >
> > Thank you for the clarification and additional results which are interesting.
> >
> > I have raised the rating after reading the rebuttal.

---

### Official Review · Reviewer_uWWw · 2023-07-03

**Soundness:** 4 excellent
**Presentation:** 4 excellent
**Contribution:** 3 good
**Rating:** 7
**Confidence:** 4

**Summary:**

The paper introduced a novel inference strategy for transformer models that focuses on inference efficiency. Instead of retaining all context tokens throughout the entire inference process, they gradually eliminate tokens as they move deeper into the layers. To determine which tokens to drop, they trained some small linear layers to predict the remaining relevant context. Their experiments revealed that approximately 80% of the tokens could be safely discarded without much adverse impact on downstream task performance or perplexity. As a result, this approach significantly reduces the computational resources required for inference when the context length exceeds 500 tokens.

**Strengths:**

- The paper demonstrates excellent writing with a clear flow of ideas.
- The authors introduce a unique and innovative data structure that enables batch operations involving masked tokens.
- The experimental results indicate that in scenarios where long context (> 500 tokens) is involved in the inference process, the models can safely drop up to 80% of the tokens without any impact on perplexity.

**Weaknesses:**

- **The choice of downstream tasks raises questions:** The downstream tasks evaluated in Figure 5 primarily involve small context sizes, and Figure 7 indicates that a smaller context leads to reduced throughput compared to the standard dense model. It would be more persuasive to demonstrate that task performance remains intact when longer contexts are required, while simultaneously achieving gains in inference efficiency.
- **Insufficient experiments with stronger base models:** It appears that the proposed method inevitably leads to performance degradation for larger and more capable base models, as evident from both Figure 5 and 7. This is likely because stronger base models are better at utilizing contexts, and additional contextual information enhances performance. To strengthen the paper's argument, the authors should include more results using larger and stronger base models (>1.5B parameters), such as LLaMA, Pythia, or even OPT.
- **Evaluating generation quality:** Though the authors perform evaluations on language modeling with perplexity, it does not necessarily align with generation quality. If would be helpful if the authors could further provide evidence that dropping context tokens does not affect generation quality.

**Questions:**

- The paper introduces the use of sparse sigmoid to gradually enforce sparsity on the context tokens. It is important to understand why sparse sigmoid techniques were chosen and if they offer any specific advantages. Could other sparsity techniques, such as l0-regularization with hard-concrete distributions (Louizos et al., 2017), or techniques used in movement pruning (Sahn et al., 2020), achieve similar results?
- There seems to be a typo in Figure 5. The green curve should be labeled as "Sparse Attention".
- Figure 2 appears to be a bit confusing without proper legends. While it is clear that "X" denotes dropped tokens, the meaning of the red blocks is unclear. Could you provide an explanation or add appropriate legends?

**Limitations:**

The authors mentioned the limitation of the working being exclusively tested on autoregressive language models, and specifically GPT2 model family. The paper would be stronger if the author can show positive results on stronger base models.

---

> ### Author Rebuttal · Authors · 2023-08-08
>
> We thank the reviewer for the positive feedback and the interesting points raised. We are glad to hear that they found our method description clear and easy to follow. In what follows ahead, we would like to take the opportunity to address the primary concern.
>
> > Choice of downstream tasks and evaluation of generation quality
>
> We thank the reviewer for raising this point. We have provided more task-specific performance results on our *global response*. Specific tasks selected, better capture long-range dependencies and interactions across sentences and paragraphs.
>
> > Stronger base models
>
> We agree with the reviewer that generalization to stronger base models is not always straightforward. In our *global response*, we provide experimental evidence that our finetuning method works *without any adaptation* for other models, *pythia* in this case. We hope that the additional results serve as a convincing argument that the proposed method is applicable to a large class of models.
>
> > Why sparse sigmoid?
>
> We experimented with a few different ways to enforce sparsity, including $L_0$-regularization and using a sigmoid with a tunable temperature parameter [Kim et al., 2022]. We found that these techniques were unsuitable for enforcing exact sparsity, while also leading to training instabilities. The sparse sigmoid on the other hand, inspired by Peters et al. [2019], leads to exact zeros for $\alpha > 1$, while also leading to well-behaved gradients.
>
> Louizon et al. propose the hard concrete distribution, which is obtained by “stretching” a binary concrete distribution and then transforming its samples with a hard-sigmoid. They also rely on a tunable temperature parameter to control the “softness” of the distribution, similar to the techniques we experimented with, that we found unsuccessful. We do not apply our sparsity regularization in the weights directly but on the products $(\mathbf{Q}\_{\text{int}}^{\ell})\_n^T (\mathbf{K}\_{\text{int}}^{\ell})\_j$. This is where we believe the instability is emerging from, as gradients are back-propagated through the whole network.
>
> Sahn et al. select $\text{top}\_k$ entries that are active in the forward pass and use a straight-through estimator to update weights even if they are pruned in the forward pass. This pruning strategy is less dynamic, as the number of $k$ active elements is chosen in advance. In our experiments, we found that models trained with the same regularization exhibited different sparsity levels for different contexts, i.e. the model “understood” that some contexts may require more information to be preserved compared to others.
>
> Thank you for the relevant references, we will include them in the related work.
>
> > Typo in Figure 5.
>
> Thank you, we will fix this.
>
> > Legend for Figure 2.
>
> “X” denotes tokens that are currently being dropped (their cached Key-Value values still exist up to this point). Red boxes corresponds to tokens that are already dropped, for which Key-Value values are no longer being cached. We will update the legend, to make things more clear.
>
> We thank the reviewer again for the interesting points raised, we will update the paper to discuss these in detail.

---

> > ### Comment · Reviewer_uWWw · 2023-08-18
> > **Thanks for your response!**
> >
> > I have read the response and think my questions and concerns are well explained and addressed. Though I highly recommend that the authors include larger models (LLaMA-7b) to strengthen the paper, I also believe the current results sufficiently demonstrate the key arguments. Thus I will keep the original score.

---

### Official Review · Reviewer_LhSf · 2023-07-06

**Soundness:** 4 excellent
**Presentation:** 4 excellent
**Contribution:** 3 good
**Rating:** 7
**Confidence:** 4

**Summary:**

The authors propose a modified dynamic masking operation to the traditional multi-headed attention in transformers in order to allow models to learn to drop tokens at specific layers during training. In order to facilitate learning, they use a sparse sigmoid that is annealed to interpolate from a traditional sigmoid up to a step function in the infinite limit. They show that this dynamic sparse attention achieves lower perplexity at the same sparseness levels compared to other modified attention mechanisms, as well as better zero-shot accuracy and throughput. The proposed model shows promise for improving the efficiency and overcoming the quadratic time complexity of traditional dense transformers.

**Strengths:**

Improving the efficiency of vanilla dense transformers is an important problem, and being able to learn the sparsification operation rather than setting a static prior is an interesting direction. The proposed method is simple and computationally efficient, and the manuscript is well written and clear. The results demonstrate marked improvements over other sparse attention mechanisms at similar sparsity levels as well as increased throughput.

**Weaknesses:**

The main concerns I have are with how well this adaptive sparsity can be used when training from scratch and on long range dependency benchmarks such as Long Range Arena [1]. Since the experiments are initialized from GPT-2, all dense information is present, the model simply has to learn which information it can safely ignore. However, when training from scratch this optimization problem becomes much more difficult since falling into a local optimum with respect to pruning early on can reduce the model’s effectiveness in the future as the $\alpha$ annealing increases towards a step function. It would be interesting to see results (even on small models) on how much utilizing this mechanism during from scratch training affects perplexity, given the same compute budget.

[1]  Long Range Arena: A Benchmark for Efficient Transformers. 2020. Yi Tay, Mostafa Dehghani, Samira Abnar, Yikang Shen, Dara Bahri, Philip Pham, Jinfeng Rao, Liu Yang, Sebastian Ruder, Donald Metzler

**Questions:**

- The $\alpha$-sigmoid operator is a bit confusing to me. How are argmax values calculated efficiently and how are gradients propagated through the argmax operator in equation 8?
- Do the baseline GPT-2 models use FlashAttention in their implementation? If not, how does the throughput of the dynamically pruned model compare to one with FlashAttention?
- How sensitive is the training to the correct annealing of the $\alpha$ parameter?

**Limitations:**

Yes

---

> ### Author Rebuttal · Authors · 2023-08-08
>
> We thank the reviewer for recognizing the importance of efficient inference and the novelty of dynamically pruning the context as a successful alternative. In the following, we take the opportunity to address the comments made.
>
> > How well this adaptive sparsity can be used when training from scratch
>
> Please see also our *global response*. We do not train from scratch to avoid extra computational costs. Training from scratch is a possibility, and local minima can be avoided by using a schedule for the sparsity parameter. However, our method is **designed** for finetuning, which is the most compelling option.
>
> > Long range dependency benchmarks
>
> We agree with the reviewer that a close examination needs to be performed to determine whether long-range dependencies can be preserved. See our *global response* for some additional results on benchmarks that better capture some of these interactions. Long Range Arena represents a suitable and attractive benchmark for encoder models, where computational benefits from sparsity are minimal. This is due to the lack of the possibility for a cumulative effect in dropping, as the one we have introduced through Eq. (6). We could not find a suitable benchmark for autoregressive models. As longer contexts become more relevant, however, we are sure that such benchmarks will emerge, as hinted by concurrent work [1].
>
> [1] ​​Mohtashami, Amirkeivan, and Martin Jaggi. "Landmark Attention: Random-Access Infinite Context Length for Transformers." arXiv preprint arXiv:2305.16300 (2023).
>
> > How are gradients propagated through the argmax operator in equation 8?
>
> The forward pass is calculated efficiently via a bisection approach, by iteratively narrowing down the interval that contains the exact solution. The backward pass can then be computed independently. For details on the backward pass, we refer to Section 3.4 of [2]. As this is essential to our work, we will update the text with a concrete pointer to the section of the cited paper.
>
> [2] Peters, Ben, Vlad Niculae, and André FT Martins. "Sparse sequence-to-sequence models." arXiv preprint arXiv:1905.05702 (2019).
>
> > Do the baseline GPT-2 models use FlashAttention in their implementation?
>
> For all attention operations, we use flash-attention as provided by the `scaled_dot_product_attention` in *pytorch-2.02* (L462-463 in Appendix). Flash attention becomes increasingly more important as sequence length increases and so removing flash attention will further highlight gain with respect to the baselines. Thank you for pointing this out, we will highlight this in the main text.
>
> > How sensitive is the training to the correct annealing of the parameter?
>
> We provide some results in Appendix B and more specifically in Figure 10. In short, increasing $\alpha$ rapidly does not allow for the new interaction parameters to be properly tuned, i.e. the achieved sparsity is worse for the same perplexity. Increasing it too slowly leads to solutions of equal quality, but requires more compute for the finetuning phase.
>
>
> We thank the reviewer again for all the questions, we will update the paper to highlight the points raised.

---

> > ### Comment · Reviewer_LhSf · 2023-08-18
> >
> > Thanks for the response. My questions have been answered but I will keep my score.

---

### Official Review · Reviewer_65eF · 2023-07-07

**Soundness:** 4 excellent
**Presentation:** 3 good
**Contribution:** 3 good
**Rating:** 7
**Confidence:** 4

**Summary:**

Given the trend in large language models, it is a pretty important problem to search to efficient architectures. In this direction is the line of work to make the attention component efficient by introducing sparsity in the attention block and allowing every token to attend only a subset of the previous tokens.

The paper presents usage of Adaptively Sparse Attention whereby
1) The network learns to drop parts of the context no longer required.
2) The tokens dropped at every layer are independent, and different set of tokens might be chosen to be persisted at every layer.
3) The network is trined using sparse sigmoid functions that introduce sparsity during training itself in contrast to some works that have tried to introduce sparsity only during inference.

Experiments show that this methodology can lead to pretty strong performance (with minimal perplexity loss), even with 80% sparsity.

**Strengths:**

The work presented in the paper is pretty innovative and impactful in this direction as finding efficient transformer architectures is key to sustain and grow the research and production usage of these large language models.

The model successfully exploits the fact that the attention matrix in these models are pretty sparse and additionally encourages that with sparse sigmoid-like function to mask out some tokens.

Experiments show pretty strong performance even with 80% sparsity in the network.

**Weaknesses:**

Given that the model is performing pretty well in terms of perplexity even with huge sparsity, it would be interesting to perform an analysis of whether there is a class of NLP tasks the is significantly affected (like knowledge intensive tasks?), or perhaps translation or similar tasks where the structure of the input sentence matters if the model is focusing to forget stopwords very early.


**Questions:**

- Was there any analysis done on which set of NLP tasks get affected the most with such sparsity?
- For inference on a given device, do we have numbers on the increase in the max sequence length that can be supported?
- Do these changes impact the training time throughput in any way?


**Limitations:**

None discussed in paper, and nothing important that I can think of.

---

> ### Author Rebuttal · Authors · 2023-08-08
>
> We thank the reviewer for the positive feedback and for acknowledging the impact of efficient inference in autoregressive models. Here, we discuss the comments raised.
>
> > Was there any analysis done on which set of NLP tasks get affected the most with such sparsity?
>
> Such an analysis is indeed interesting and would enhance our intuition. We have provided some preliminary evidence (see our *global response*) on task-specific performance and on zeroshot performance on some additional tasks that we consider interesting in the high-sparsity regime. Thank you for raising this point, we will update the paper with a concrete discussion on this matter.
>
> > For inference on a given device, do we have numbers on the increase in the max sequence length that can be supported?
>
> We agree with the reviewer that increasing the maximum sequence length during inference for a given device is one of the most promising outcomes of pruning the context. In our case, the maximum context length is limited by the maximum supported sequence length of the pre-trained GPT model, see also our *global response*. Provided a base model that supports larger sequence lengths, benefits given fixed hardware, are substantial (see Table 1 of attached PDF in our *global response*). We also present results for *pythia* models that already support longer context windows (see Figure 2 of the attached PDF in our *global response*).
>
> > Do these changes impact the training time throughput in any way?
>
> See also our *global response*. Training FLOPs are only marginally affected. Still, we only finetune pre-trained models, to avoid even the smallest increase in training throughput.
>
> We thank the reviewer again for all the questions. We truly believe that addressing them helped improve our work significantly.

---

> > ### Comment · Reviewer_65eF · 2023-08-16
> >
> > Thanks for the clarifications! Nice work.
> >
> > I have read through the responses and will keep my original scores.

---

### Author Rebuttal · Authors · 2023-08-08

We would like to thank all reviewers for taking the time to review our paper and for the valuable feedback. Here we address common points raised and present more experiments that we believe further strengthen our findings.

**How sparsity affects specific NLP tasks (@65eF, @LhSf, @uWWw)**

Evaluating GPT models pretrained by language modelling objective is in general *challenging* and different techniques have been developed based on the final intended use. This challenge motivated us to showcase zeroshot accuracy to downstream tasks, in addition to evaluating perplexity for upstream tasks. The tasks we selected are commonly used, see e.g. Dettmers et al. [2022], Frantar et al. [2023b] or other popular benchmarks [gpt4all](https://gpt4all.io/index.html), [ilm-eval](https://tju01.github.io/ilm-eval/#?benchmark=lm-evaluation-harness). We agree with the reviewers, however, that some tasks may be affected more by sparsity compared to others. For that reason, we provide per-task zeroshot performance in the attached PDF, see Figure 1. We also include zeroshot results for additional tasks, that require long-range dependencies. It becomes clear that some tasks are affected more than others, according to our intuition.

**Impact on training time (@65eF, @LhSf)**

The dropping mechanism introduces a *marginal* computational cost in terms of FLOPs, as highlighted in Figure 6 (left). Furthermore, our approach can be applied as a post-processing step given an initial model trained in a standard fashion, decreasing the training cost even further. Training from scratch is also a possibility, given a suitable decay schedule for the sparsity parameter, but fine-tuning is more compelling as it can be applied to existing pretrained models. Also note that one could tune our hyperparameter $\gamma$ during training dynamically, to achieve a desired level of sparsity, e.g. by monitoring the running average of the sparsity during training and increasing/decreasing it in small steps (similar to ADA [1] in GANs). As, for evaluation purposes, we wanted to acquire multiple models with different levels of sparsity, we did not do that.

**Additional Models and Inference Compute (@65eF, @uWWw, @jYJo)**

As the original context grows, pruning the context makes an increasingly significant difference, as more opportunities for pruning exist. This is clearly demonstrated by Figure 7 (bottom right). Generally, enlarging the context length of Transformers is a fascinating area of research with novel techniques being currently proposed, e.g. [2, 3]. In our experiments, we are limited by the maximum positional encodings of the base GPT models. To evaluate how our method performs in larger contexts, we additionally present preliminary results on *pythia* language models. *Pythia* uses rotary position embedding [4], as many current state-of-the-art LLMs, e.g. LLaMa-(1 and 2).
We present results on “Perplexity vs sparsity” in the attached PDF, in Figure 2. These concrete results demonstrate that *our pruning technique works out of the box for different positional encodings* as well. Longer contexts supported by our base model (for pythia models that is 2048) additionally allow for higher levels of sparsity for the same performance, in the case of Figure 2 in the attached PDF, that is perplexity. In the future, we intend to also evaluate larger models, potentially with the use of LoRA [5]. We again highlight that our approach is not specific to a particular language model, as it can be applied to any autoregressive transformer architecture.

An additional benefit of our pruning strategy is that we can accommodate longer initial contexts for the same hardware, as raised by reviewer **@65eF**. Table 1 in the attached PDF showcases the maximum potential context windows for different batch sizes. Our strategy accommodates inference with much longer contexts, for a fixed device.


[1] Karras, Tero, et al. "Training generative adversarial networks with limited data." Advances in neural information processing systems 33 (2020): 12104-12114.

[2] Press, Ofir, Noah A. Smith, and Mike Lewis. "Train short, test long: Attention with linear biases enables input length extrapolation." arXiv preprint arXiv:2108.12409 (2021).

[3] Chen, Shouyuan, et al. "Extending context window of large language models via positional interpolation." arXiv preprint arXiv:2306.15595 (2023).

[4] Su, Jianlin, et al. "Roformer: Enhanced transformer with rotary position embedding." arXiv preprint arXiv:2104.09864 (2021).

[5] Hu, Edward J., et al. "Lora: Low-rank adaptation of large language models." arXiv preprint arXiv:2106.09685 (2021).

---

### Decision · Program_Chairs · 2023-09-21

**Decision:**

Accept (spotlight)

**Comment:**

This paper presents an efficient and interpretable context-based dynamic pruning method for transformers, which allows models to learn to drop tokens at specific layers during training. The authors introduce a sparse sigmoid and regularization term to control the sparsity and use additional query/key layers to generate dynamic attention masks and sparsify self-attention maps. The experiments demonstrate that the proposed method achieves lower perplexity at the same sparseness levels compared to other modified attention mechanisms, as well as better zero-shot accuracy and throughput. The results also show that in scenarios where long context (> 500 tokens) is involved in the inference process, the models can safely drop up to 80% of the tokens with little impact on perplexity.

The reviewers found the paper to be well-written, clear, and innovative. The proposed method is well-motivated and easy to follow, and the authors provide an extensive analysis of the relationship between context length, throughput, and speed in various parameter sizes. They also devise efficient batched data structures for optimized computation based on the proposed pruning method.

However, some concerns were raised regarding the choice of downstream tasks, the performance of the method when training from scratch, and the evaluation of generation quality. It would be interesting to see results on long-range dependency benchmarks and stronger base models (>1.5B parameters), as well as further qualitative studies of interpretability varying the sparsity level.

Overall, the paper is technically solid and has high impact on NLP. The reviewers recommend acceptance of the paper.